# Combined Inclusion of Former Foodstuff and Distiller Grains in Dairy Cows Ration: Effect on Milk Production, Rumen Environment, and Fiber Digestibility

**DOI:** 10.3390/ani12243519

**Published:** 2022-12-13

**Authors:** Ludovica Maria Eugenia Mammi, Giovanni Buonaiuto, Francesca Ghiaccio, Damiano Cavallini, Alberto Palmonari, Isa Fusaro, Valentina Massa, Andrea Giorgino, Andrea Formigoni

**Affiliations:** 1Department of Veterinary Medical Sciences, Alma Mater Studiorum—University of Bologna, 40064 Bologna, Italy; 2Faculty of Veterinary Medicine, University of Teramo, 64100 Teramo, Italy; 3Dalma Mangimi S.p.a., 12030 Marene, Italy

**Keywords:** former foodstuff, wheat distillers’ grain, livestock sustainability, circular economy, milk quality, dairy cows

## Abstract

**Simple Summary:**

One-third of the global food produced for human consumption is wasted every year. This leads to the wasting of economic, environmental, and social resources. The reallocation of some unconventional agro-industrial by-products, such as former foods or distiller grains, into the animal feed chain, can contribute to increasing the sustainability of livestock production, reducing the need for natural resources such as arable soil, water, fertilisers, and fuels, and consequentially reduce the impact of animal requirements. Many agro-food industrial by-products are already used in animal feeding, but the potential of these feed ingredients has not yet been fully investigated, especially in ruminant nutrition.

**Abstract:**

The aim of the present study was to investigate the effect of the substitution, in dairy cow rations, of traditional protein and starch sources with more sustainable “circular” feeds to increase the sustainability of dairy production. For this purpose, eight multiparous mid-lactating cows were blocked and assigned to one of four treatments and were used in a replicated 4 × 4 Latin squares design with 21-days periods (14 days of adaptation and 7 of data collection). Two different circular feedstuffs were tested: a bakery’s former foodstuff (FF) and a wheat distiller’s grain with solubles (WDGS). These ingredients were used, alone and in combination, in three experimental diets (FF, WDGS; FF + WDGS) and compared to a standard ration (CTR). Dry matter intake and rumination time were not influenced by these diets. Conversely, dietary treatments partially influenced the milk yield, rumen pH, Volatile Fatty Acids (VFA) production, and fibre digestibility. In particular, the combined inclusion of FF and WDGS increased milk production (37.39 vs. 36.92, 35.48, 35.71 kg/day, for FF, WDGS and CTR diets, respectively) and reduced milk urea content (13.14 vs. 16.19, 15.58, 16.95 mg/dL for FF, WDGS, and CTR diets, respectively). No effects of this association were found in the milk composition, acetic and propionic production, and fibre digestibility. These results suggest that the association of former foodstuff and wheat distillers’ grains could be safely included in dairy cow rations to increase the sustainability of cow nutrition and improve milk production without impairing animal health, dry matter intake, and fibre digestibility.

## 1. Introduction

Animal feeding in livestock production represents, depending on the species, about 60–85% of the total production cost [1]. Therefore, adopting proper feeding strategies will become increasingly important to improve the efficiency of livestock systems. In this scenario, the use of innovative or alternative feed ingredients in farm animal diets can be an interesting option to improve the rational use of natural resources, such as soil and water, and increase the sustainability of animal production [2,3]. An interesting perspective is related to the ability of livestock to turn human food by-products (also called ex-food or former foodstuffs; FFPs) into high-biological value animal feed [4,5,6]. According to Fausto-Castro et al. [7], from a circular economy perspective, feeding animals with FFPs is a useful option to save natural resources and reduce food waste.

The industries that process cereals for human consumption producing pasta, snacks, bakery, and pastry products, make available significant quantities (about 3–4% of the total) of products not placed on the market due to manufacturing and packaging defects which, however, do not imply any risk to their potential use in animal feed. The European Former Foodstuff Processors Association (EFFPA) estimates that 5 Mt of FFPs is yearly processed into animal feed in the EU. According to Luciano et al. [8], FFPs include leftovers from the bakery industry (such as bread or pasta) and leftovers from the confectionery industry (i.e., chocolates, biscuits, etc.). Overall, FFPs have a high content of starch, oils or fats, and sugar [9] and, therefore, a great energy content. Bread, salty cakes, and snacks, thanks to the long cooking process, are easily digestible and represent a good source of energy due to their high digestible starch content. Differently, confectionary products (for example, chocolates or biscuits) can be considered supplementary feeds, available all year round and rich in sugars, fat, and energy. The feed sector provides for a combination of different technological processes (i.e., unpacking, mixing, grinding, pelleting, and drying) for the processing of FFPs as feed ingredients that can be used as substitutes for various raw materials in compound feeds [10,11,12]. These products are already used in animal feeding, especially in the pig and poultry industry [13,14], in compliance with EU regulations on feed [15], but are still in a limited amount (3.3% out of the total food waste). The use of these ingredients in ruminant nutrition is particularly limited, especially in dairy cows, because of the particular composition of these products and the lack of information regarding their effects on the rumen environment and quality of production. Few studies, to our knowledge, explored the effects of former foodstuffs on dairy cows’ performances and health [16,17]. Kaltenegger et al. [17] substituted up to 30% of traditional cereals with feed made by bakery former food products, highlighting the significant but not detrimental effects on rumen pH and volatile fatty acids (VFA), fibre degradability, microbial community, milk production, and the metabolic profile [16,17]. On the contrary, in their in vitro study, Hummer and colleagues found that higher levels of substitution (45%) had negative effects on the same parameters evaluated by in vitro fermentations [16].

Co-products such as distillers’ grain with solubles (DGS) from corn or wheat in the alcohol industry have been extensively investigated in dairy cow feeding [18,19,20,21,22,23], and these products are considered not detrimental to the cows’ performances and health at 20% of Dry Matter (DM) inclusion [20]. However, to our knowledge, the effect of the combined inclusion of these circular feeds on dairy cow health and production has not been investigated yet.

The association of DGS and FF could serve as a replacement for both starch and protein sources, thus, substantially increasing the sustainability of dairy farming by reducing land exploitation for cereals and legumes production.

Therefore, the aim of the present study was to investigate how the combined use of bakery former food products and wheat distillers’ grain with solubles influence milk production, rumen environment, and fibre digestibility in lactating dairy cows.

## 2. Materials and Methods

The experiment was carried out between September and December at the experimental dairy farm of the Department of Veterinary Medical Science (DIMEVET) of the University of Bologna, located in Ozzano dell’ Emilia, province of Bologna (Bologna, Italy). All the experimental procedures were conducted in compliance with the European Directive on the Protection of Animals Used for Scientific Purposes [24].

### 2.1. Animals and Experimental Design

Eight multiparous Holstein cows between 60 and 110 Days In Milk (DIM) (average Body Weight (BW): 588 ± 50 kg, mean ± standard deviation) were blocked by parity (1.63 ± 0.64 days, mean ± SD) and milk production (40.68 ± 3.94 kg/day, mean ± SD), and were enrolled in a replicated 4 × 4 Latin square design study with four 21-days periods (14 days of adaptation and 7 days of data collection). The dietary treatments differed for the starch and protein sources that were partially replaced with sustainable feeds coming from the grain industry: a condensed wheat wet distiller grain with solubles (WDGS) and a pelleted FFPs (FF) provided by a feed company that specialized in the recycle of bakery industry waste (Dalma Spa, Marene, Italy). The composition of the experimental feeds (CTR, WDGS, and FF) are reported in Table 1. The four diets (Table 2) included traditional starch and protein sources in the control diet (CTR), the wheat wet distiller grain (WDGS diet, 4 kg/day), the pelleted FFPs feed (FF diet, 3 kg/day), or a combination of both (WDGS + FF diet, 4 + 3 kg/day). FFPs feeds are composed of bakery industry wastes such as pasta, bread, biscuits, and snacks which are no longer intended for human consumption [25].

Cows selected for the research were housed in a naturally ventilated tie-stall barn with individual feed managers and drinkers, equipped with a weighing scale and water meter for the daily recording of feed and water intake. Rations were offered in individual feed managers as total mixed rations (TMR) once a day ad libitum allowing 10% orts. The diets were formulated using DinaMilk5 (Fabermatica, Ostiano, Italy) to mimic TMR used in the Parmigiano Reggiano cheese production area of Italy and consisted of all dry and nonfermented components [26].

Cows were milked twice a day (08:00 a.m. and 07:30 p.m.) in a double-5 herringbone milking parlour equipped with a system to record the daily individual milk production and body weight (Afimilk Information Management System, Afikim, Israel).

### 2.2. Samples and Data Collection

Data and samples were collected during the experimental weeks, from day 15 to 21 of each period. The individual feed and water intake were recorded daily. Dry matter intake (DMI) was determined by recording daily feed offered and refusals.

The feedstuffs were sampled at the beginning of each period, while TMR and orts samples were collected 3 times/period. The nutritional value and characteristics of FF and WDGS were determined before the beginning of the trial for the correct formulation of the experimental diets (Table 1 and Table 2). All samples were analysed in the laboratories of the Animal Production and Food Safety Service (SPASA; DIMEVET, University of Bologna, Ozzano Emilia, Italy).

Feed and TMR samples were dried in a forced-air oven at 65 °C for 48 h for DM determination and were analysed by wet chemistry according to the following procedures: crude protein (CP) according to AOAC [27] (method 976.06 and 984.13) using a Kjeldahl nitrogen analyser (Gerhadt Vapodest 50, Gerhardt GmbH, Königswinter, Germany), starch determined according to Ehrman [28] and AOAC [27] (method 920.40), ether extract according to AOAC [27] (method 920.390020), ash-corrected α-amylase–treated neutral detergent fibre (NDF) with the addition of sodium sulfite (aNDFom), acid detergent fibre (ADF) and acid detergent lignin (ADL) according to Mertens et al. [29], undigested NDF (uNDF) according to Cotanch et al. [30] AOAC [27] (method 973.18) and ash after 4 h of combustion in a muffle furnace 550 °C (Vulcan 3–550, Dentsply Neytech, Burlington, NJ, USA). The starch content of WDGS was determined by polarimetric methods, while the lipids were extracted with petroleum after hydrolysis with HCl, as described in the commission regulation [31]. The total sugars and fractions (sucrose, glucose, and fructose) were determined using an enzymatic method (Megazyme International Ltd., Bray, Ireland), as described by Palmonari et al. [32]. The starch digestibility of the former foodstuff feed was evaluated by 7 h in vitro fermentation as described by Ferraretto et al. [33].

Wet chemistry results were used to calibrate a near-infrared (NIR) instrument fitted with a spinning cup holder (NIRSystem 6500; Perstorp Analytical Inc., Silver Spring, MD, USA) according to Giaretta et al. [34] and to Buonaiuto et al. [35].

Individual milk production was recorded daily in the milking parlour. During the experimental week, individual milk was sampled from 2 consecutive milkings and was analysed for fat, protein, lactose, somatic cells (SCC), and urea by infrared spectroscopy procedures (Foss 4000, Foss, Hilleroed, Denmark) by a certified lab (Granarolo spa, Bologna, Italy). Fat-corrected milk (FCM 4%) and energy-corrected milk (ECM 4%) were calculated according to Davidson et al. [36] with the formula:ECM = (milk yield × 0.327) + (milk fat × 12.86) + (milk protein × 7.65)FCM = (milk yield × 0.4324) + (milk fat × 16.2162)

According to Cavallini et al. [37], feed efficiency during the experimental weeks was calculated as milk/DMI (kg/kg).

Fecal samples were collected 2 times from each cow on days 19 and 20 of each period at 8 a.m. and 8 p.m. Faecal composition (moisture, starch, protein, lipids, ash, and fibre fractions) and fibre digestibility at 24 (pdNDF24) and 240 h (pdNDF240) were analysed by NIRs to evaluate the total tract fibre degradability (TTDpdNDF) of the diets [38]. The TTDpdNDF was calculated, according to Palmonari et al. [39], with the following formula:TTDpdNDF (% pdNDF) =100 − (uNDFdiet/uNDFfeces) × (pdNDFfeces/pdNDFdiet) × 100
where TTDpdNDF (% pdNDF) is the total-tract digestibility of potentially digestible NDF, uNDF is the unavailable NDF, and pdNDF is the potentially digestible NDF.

Rumination time (min/day) was recorded continuously by collars (Heat Time Pro, SCR, Netanya, Israel) [40], as well as rumen pH, which was recorded every 10 min by indwelling wireless probes (SmaXtec Animal Care Sales GmbH, Graz, Austria), previously validated [41]. On day 20 of each period, a sample of the rumen fluid was collected from each cow by an esophageal probe to evaluate the content of ammonia (NH3) and volatile fatty acids (VFA). Ammonia was determined by a commercial kit (urea/BUN—colour, BioSystems S.A. Barcelona, Spain) and VFA by gas chromatography [42]. The analysed VFAs were: acetic, propionic, iso-butyric, butyric, isovaleric, and valeric acid. VFAs were separated using a Fisons HRGC MEGA 2 series 8560 (Fison Instruments, Glasgow, UK) with a flame ionization detector (Fison Instruments, Glasgow, UK)) on a 2 m glass column (inner diameter, 3 mm) of 10% SP-1000 + 1% H_3_PO_4_ on 100/120 Chromosorb W AW (Tehnokroma Analitica S.A., Sant Cugat del Vallès, Spain) with nitrogen as the carrier gas. The temperature of the injector and detector was 200 °C, and the oven temperature was 155 °C. The internal standard adopted was 2-ethylbutyric acid (Sigma Aldrich, Taufkirchen, Germany).

### 2.3. Statistical Analysis

Statistical analysis was performed using the software JMP Pro (v15, Statistical Analysis Systems Institute Inc., Cary, NC, USA). First, the normal distribution of data was tested by a Shapiro–Wilk test, and according to that, all the data were analysed by a mixed model as a replicated Latin square design. The diet (CTR, FF, WDGS, or WDGS + FF), period and interactions were considered fixed effects and the cow within the replicate was considered random. Continuous data, such as DMI, rumination time, r-pH, and milk production, were treated as repeated measures, with the cow as the subject. When a significant F test (*p* < 0.05) was detected, a means multiple comparisons were performed by Student t-test.

## 3. Results

Results about the dry matter and water intake; the rumination time and rumen pH are reported in Table 3.

The dry matter intake (DMI) of cows was, on average, 22.87 ± 3.13 kg/day and did not differ between the dietary treatments (*p* > 0.05), similar to the water intake that was, on average, 137.4 ± 22.05 L/day. The daily time spent ruminating did not change with the different diets, remaining always around physiological levels (mean ± s.d, 517 ± 70.45, min/day). Conversely, the diets partially affected reticolo-rumen pH (r-pH) (Table 3). In particular, the daily average pH was similar between the different diets (CTR: 6.13, WDGS: 6.13, FF: 6.12, WDGS + FF: 6.10, SEM 0.03, *p* > 0.05), while the time below pH 5.8 was longer with the WDGS + FF diet (WDGS + FF: 171.87 vs. CTR: 55.86, WDGS: 104.98, FF: 108.93, SEM 22.37 min/day, *p* < 0.05), and tended to be longer in the FF diet when compared to the control (108.93 vs. 55.86, *p* < 0.1). Similarly, the time with a pH below 5.5 tended to be longer in the WDGS + FF diet (WDGS + FF 27.16 vs. CTR 11.65, WDGS 14.49, FF 19.63, SEM 1.37 min/day, *p* < 0.1). The results of ammonia and VFA content in the rumen are reported in Table 4. In particular, the concentration of ammonia (mg/100 mL), total VFA, and acetic acid (mmol/L) was lower in the WDGS diet compared to the CTR and FF diets (*p* < 0.05). Conversely, the FF diet showed a higher rumen concentration of propionic acid compared to the WDGS one (*p* < 0.05). However, despite the differences in VFA concentration, the molar proportion of the different VFA remained constant between the diets.

Considering milk production, the best productive performances (Table 5) were achieved with the WDGS + FF diet, in which the daily milk yield was significantly higher (*p* < 0.05) than the CTR and WDG diets. The FF diet showed intermediate results that were statistically similar to the WDGS + FF diet and the control, while the WDGS showed the lowest milk yield and ECM (*p* < 0.05) among all the diets employed.

However, feed efficiency, calculated as the milk (kg/day)/DMI (kg/day) and fat-corrected milk (FCM), were not influenced by the dietary treatments (*p* > 0.05).

Similarly, the milk fat and lactose percentage remained unvaried between the diets, while milk protein and urea content showed some differences (*p* < 0.05, Table 5). In particular, milk protein (%) was lower in the WDGS (3.35%) and FF (3.33%) diets compared to the others (CTR: 3.44; WDGS + FF: 3.36%), while the milk urea content was lower in the WDGS + FF (13.14 mg/100 mL).

The data on the faecal composition and fibre degradability (TTDpdNDF) are reported in Table 6. The faecal composition remained similar with the different diets, except for the ADL content that increased significantly with the WDGS diet (*p* < 0.05). Conversely, the WDGS diet had the lowest value of TTDpdNDF (*p* < 0.05).

## 4. Discussion

This study aimed to investigate the effects of the inclusion, both alone and in combination, of two bakeries and agro-industrial by-products (FF and WDGS) on milk production and composition, rumen environment, and fibre degradability. Previous studies have investigated the effects of WDGS inclusion or FF in dairy cow rations [17,18,20], but, to our knowledge, this is the first one to evaluate the effects of the inclusion of both ingredients together.

In particular, the effects of DGS inclusion have been fully investigated in dairy cows [43,44,45,46], even though most of these studies refer to corn DGS [43,44,46], and only a few to wheat DGS, such as the one used in our study [47,48]. The different plant origins and producing technologies significantly influence the chemical composition and the nutritive value of the product [48]. Indeed, the WDGS used in the present study, compared to the study of De Boever et al. [48], had a higher content of sugars (28.41% vs. 12.3% DM) and DM (35.48 vs. 27.8%) and a lower content of CP (22.12 and 32.5% DM).

Similarly, the FFPs composition was extremely variable and depended on several factors, such as the type of products from which they were obtained, their origin (factory, region), storage, and processing methods. In general, FFPs deriving from confectionary products (such as biscuits, chocolate, or dry cakes) have a high content of simple sugars and fats, while those coming from the bakery industry are characterized by high levels of high digestible starch. For these reasons, the capability of the processor to obtain homogeneous products starting from different ingredients is extremely important for the farmers that need to have over time the constant nutritional characteristics of FFPs.

The FF used in the present study had similar characteristics to those reported previously [8,9], characterized by high levels of fat (10% DM), starch (44% DM), protein (10% DM), and sugars (11% DM).

In our study, the inclusion of FF and WDGS, alone or in combination, in dairy cow rations did not influence the dry matter or water intake (Table 3), confirming the absence of negative effects in these ingredients on the palatability of the administrated TMR. This result is in line with those reported by Karlsson et al. [49] with a by-product-based concentrate and by Pinotti et al. [9], who revised the literature concerning the use of bakery by-products in monogastric and ruminants’ nutrition.

Most important, in our study, the inclusion of these feeds had no detrimental effects on rumen physiology, as confirmed by the analysis of the rumen environment, including rumination time, rumen pH, and the concentration of ammonia and VFA.

Indeed, no significant differences were observed between the rumination time and average daily pH in the different diets, while the combination of WDGS and FF resulted in the longest time with a pH lower than 5.8 (Table 3) and induced a trend towards a slightly longer time with pH below 5.5 compared to the CTR diet. However, the time recorded either with a pH below 5.8 or below 5.5 remained within the normal physiological ranges of the cows for all diets and was far from the values indicative of subacute ruminal acidosis [50]. Rumen pH, ruminal fluid VFA concentration was only partially affected by the dietary treatments, while the molar proportion of different VFA remained constant. The highest production of propionate recorded with the FF diet agreed with Humer et al. [16], who reported an increase in propionic acid in diets containing bakery by-products without unfavourable effects on rumen pH by up to 30% of the DM level of inclusion. Conversely, the lower production of acetic and propionic acids recorded with the WDGS diet was unexpected, as well as the reduced fibre digestibility recorded with this diet. Indeed, to our knowledge, no other studies reported similar results, and on the contrary, an increase in propionic acid proportion [22,45,47] and no effects on fibre digestibility [21,22] were described in cows fed dried distillers’ grains. Interestingly, the association of the two circular feedstuff (WDGS + FF) led to an intermediate production of total the VFA without altering their molar proportion, suggesting a positive interaction of the two ingredients on the rumen environment.

In addition, the combined inclusion of FF and WDGS produced the best milk performance in the cows that showed higher milk yield without alterations in milk components and with a reduced content of urea. This result is extremely important because, combined with the increased milk production, this suggests that this diet permits better nitrogen utilization by the animals.

While the association of this kind of by-product has never been studied before, an increased milk production when feeding bakery FFPs was previously reported by other authors [17,51].

On the contrary, the WDGS diet had a productive performance similar to the CTR but worse than the other two experimental diets (FF and WDGS + FF). These results only partially agree with the previous literature, which reports that feeding distiller grains with solubles to dairy cows either improved [43,44,45,46] or had no effect on milk production parameters [23].

## 5. Conclusions

The results of the present study suggest that the inclusion of WDGS and FF, alone or in combination, in properly formulated diets had no negative effects on dairy cows’ health and productivity. Most of all, the present research shows a positive interaction between the two by-products, particularly on milk production, rumen environment, and nitrogen efficiency. These results are of great interest from an environmental point of view.

Indeed, the possibility to safely combine in the same ration two circular feeds with different nutritional profiles, offers the opportunity to fully meet the requirements of high-producing dairy cows, while reducing nitrogen excretions and increasing the circularity of the system. So far, it can be stated that the rational inclusion of feeds based on former foodstuff and wheat distillers, in properly formulated rations, represents a safe opportunity to reduce the environmental impact of dairy farming while maintaining high levels of production.

Despite previous studies confirmed results on the productive performance and rumen fermentations of these ingredients when fed alone, data about the effects of these ingredients, alone or in combination, on the cheesemaking properties of milk, and the final characteristic of cheese are still lacking. Therefore, further research in this area is encouraged in order to promote the use of these sustainable feeds in cheese industries, thus fostering the transition of the entire dairy chain towards a circular farming system.

## Figures and Tables

**Table 1 animals-12-03519-t001:** Chemical composition ^1^ of the different preparations tested ^2^.

Component, % DM	WDGS	FF
DM, % as fed	35.48	90.33
Moisture	64.52	9.68
CP	22.12	10.57
EE	4.29	10.11
Starch	17.14	44.12
aNDFom	1.10	13.24
ADF	1.09	6.49
ADL	0.27	3.01
Ash	11.44	2.16
Sugars ^3^	28.41	11.3
Sucrose	2.16	0.3
Glucose	4.44	7.5
Fructose	4.30	2.5
Starch digestibility ^4^, %	--	87.36
pH	3.47	--

^1^ DM—Dry matter; CP—Crude protein; EE—Ether extract; aNDFom—amylase-treated ash-corrected NDF with addition of sodium sulphite; ADF—Acid detergent fiber; ADL—Acid detergent lignin ^2^ WDGS—condensed wheat distiller soluble; FF—former foodstuff. ^3^ Total sugars determined by enzymatic method (Megazyme International Ltd., Bray, Ireland). ^4^ In vitro starch digestibility at 7 h.

**Table 2 animals-12-03519-t002:** Ingredients and chemical composition of dietary treatments ^1^.

Ingredients, kg/Day as Fed	CTR	WDGS	FF	WDGS + FF
Alfa-alfa hay	8.0	8.0	8.0	8.0
Grass hay	3.0	3.0	3.0	3.0
Grain mix ^2^	7.5	7.0	7.0	5.3
Mixed flakes ^3^	7.5	7.5	5.0	5.0
Min-Vit supplement ^4^	0.7	0.7	0.7	0.7
Megafat^®^	0.5	0.5	0.5	0.5
Former foodstuff feed ^5^	--	--	3.0	3.0
Wheat wet distiller soluble	--	4.0	--	4.0
Composition, % DM				
DM, % as fed	89.7	83.1	90.1	84.3
CP	14.9	15.0	15.0	14.9
Starch	19.4	18.4	20.3	19.1
Fat	4.5	4.6	5.3	5.4
aNDFom	34.7	33.7	32.9	31.8
ADF	23.3	22.9	23.3	21.2
ADL	4.5	4.3	4.1	3.9
uNDF240	9.7	9.5	10.3	10.2
Ash	8.9	9.2	8.8	9.1
peNDFom	13.2	13.9	13.4	13.8

^1^ CTR—control; WDGS—condensed wheat distiller soluble diet; FF—former foodstuff; WDGS + FF—condensed wheat distiller soluble + former foodstuff. ^2^ Soybean meal (48% CP), 26.7%; Grain mix: corn meal, 25%; soyhulls 13.3%; corn gluten feed, 13.3%; sorghum meal, 8.7%; barley meal, 8%; cane-beet molasses blend, 5%. ^3^ Mixed flakes: corn flaked, 26.7%; sorghum flaked, 26.7%; beet pulp, 26.6%; soyhulls, 20%. ^4^ Sodium Bicarbonate, 30%; Calcium carbonate, 25.7%; dicalcium fosfate, 19.3%; sodium chloride, 16.6%; magnesium oxide, 7%; vitamines (A, D3, E), 0.2%; oligo minerals, 1.2%. ^5^ Former foodstuff feed (Dalma spa, Marene, Italy). Composed of bakery industry wastes such as pasta, bread, biscuits, and snacks no longer intended for human consumption.

**Table 3 animals-12-03519-t003:** Effects of dietary treatments ^1^ on DMI ^2^, water intake, BW ^3^, rumination time, and reticular pH.

Item	CTR	WDGS	FF	WDGS + FF	SEM ^4^	*p*-Value
DMI, kg	22.99	22.25	23.15	23.08	0.30	0.67
Water intake, L/day	136.3	140.5	137.8	135.0	5.09	0.86
BW, kg	614	619	606	607	52	0.79
Rumination, min/day	517.1	509.0	511.5	530.0	15.69	0.82
Daily average r-Ph	6.13	6.13	6.12	6.10	0.03	0.83
Time r-pH < 5.8, min/day	55.86 ^b^	104.98 ^b^	108.93 ^b^	171.87 ^a^	22.37	0.02
Time r-pH < 5.5, min/day	11.65 ^B^	14.49 ^AB^	19.63 ^AB^	27.16 ^A^	1.37	0.08

^a,b^ Values within rows with different superscripts differ (*p* ≤ 0.05). ^A,B^ Values within rows with different superscripts differ (*p* ≤ 0.1). ^1^ CTR—control; WDGS—condensed wheat distiller soluble diet; FF—former foodstuff; WDGS + FF—condensed wheat distiller soluble + former foodstuff. ^2^ DMI—dry matter intake. ^3^ BW—body weight; ^4^ Standard Error of the Mean.

**Table 4 animals-12-03519-t004:** Effects of dietary treatments ^1^ on rumen ammonia and VFA ^2^ content.

Item	CTR	WDGS	FF	WDGS + FF	SEM	*p*-Value
NH_3_, mg/100 mL	9.75 ^a^	7.72 ^b^	9.29 ^a^	8.50 ^ab^	0.805	0.02
VFA Concentration (mmol/L)						
Total VFA	95.95 ^a^	77.93 ^b^	98.63 ^a^	83.40 ^ab^	5.709	0.03
Acetic	54.89 ^a^	44.75 ^b^	53.93 ^a^	46.26 ^ab^	3.073	0.02
Propionic	26.28 ^ab^	21.08 ^b^	29.46 ^a^	24.37 ^ab^	2.586	0.03
Isobutyric	0.95 ^ab^	0.79 ^b^	1.04 ^a^	0.81 ^b^	0.078	0.03
Butyric	11.17	9.08	11.21	9.58	1.125	0.15
Isovaleric	0.93 ^ab^	0.80 ^ab^	0.96 ^a^	0.72 ^b^	0.104	0.03
Valeric	1.69 ^ab^	1.43 ^b^	1.95 ^a^	1.61 ^ab^	0.192	0.01
VFA molar proportion (% mol)						
Acetic	57.52	57.15	55.62	55.54	1.526	0.54
Propionic	27.00	27.37	28.99	29.20	1.682	0.57
Acetic/Propionic	2.22	2.18	1.98	2.01	0.230	0.69
Isobutyric	0.98	1.01	1.03	0.97	0.082	0.82
Butyric	11.74	11.60	11.43	11.44	0.644	0.96
Isovaleric	0.98	1.02	0.99	0.89	0.099	0.58
Valeric	1.77	1.86	1.95	1.96	0.135	0.38

^1^ CTR—control; WDGS—condensed wheat distiller soluble diet; FF—former foodstuff; WDGS + FF—condensed wheat distiller soluble + former foodstuff. ^2^ Volatile fatty acid. ^a,b^ Values within rows with different superscripts differ (*p* ≤ 0.05).

**Table 5 animals-12-03519-t005:** Effects of dietary treatments ^1^ on milk production and composition.

Item ^2^	CTR	WDGS	FF	WDGS + FF	SEM	*p*-Value
Milk, kg	35.71 ^cb^	35.48 ^c^	36.92 ^ab^	37.39 ^a^	0.502	0.001
ECM 4%, kg	34.94 ^ab^	33.72 ^b^	35.55 ^ab^	35.96 ^a^	0.762	0.04
FCM 4%, kg	39.05	37.74	40.00	40.37	1.150	0.23
Fat, %	3.74	3.67	3.64	3.68	0.163	0.87
Protein, %	3.44 ^a^	3.35 ^b^	3.33 ^b^	3.36 ^ab^	0.080	0.04
Lactose, %	4.79	4.82	4.77	4.81	0.033	0.73
Urea, mg/100 mL	16.95 ^a^	15.58 ^ab^	16.19 ^a^	13.14 ^b^	1.240	0.02
Milk/DMI	1.58	1.62	1.65	1.64	0.048	0.64

^a,b,c^ Values within rows with different superscripts differ (*p* ≤ 0.05). ^1^ CTR—control; WDGS—condensed wheat distiller soluble diet; FF—former foodstuff; WDGS + FF—condensed wheat distiller soluble + former foodstuff. ^2^ ECM—energy-corrected milk; FCM—fat-corrected milk; DMI—dry matter intake.

**Table 6 animals-12-03519-t006:** Effects of dietary treatments ^1^ on faecal composition and total tract fibre digestibility (TTDpdNDF).

Item ^2^	CTR	WDGS	FF	WDGS + FF	SEM	*p*-Value
Dry matter, % as fed	14.70	14.20	15.05	14.90	0.53	0.65
CP, % DM	13.29	13.32	12.70	12.86	0.38	0.42
Starch, % DM	3.90	3.56	3.76	3.59	0.14	0.23
aNDFom, % DM	57.95	59.50	58.86	58.64	1.30	0.86
ADF, % DM	43.19	44.13	43.00	43.68	0.91	0.82
ADL, % DM	16.73 ^a^	18.02 ^b^	16.47 ^a^	16.39 ^a^	0.558	0.04
Ash, % DM	10.53	10.73	10.25	10.66	0.20	0.73
uNDF240, % NDF	74.39	73.90	77.20	76.33	2.12	0.86
uNDF240, % DM	43.11	44.02	45.40	44.72	1.46	0.78
pdNDF240, % NDF	25.61	26.10	22.80	23.67	2.12	0.64
pdNDF240, % DM	14.84	15.49	13.46	13.91	1.34	0.91
pdNDF24, % NDF	15.22	15.37	15.46	15.20	0.30	0.62
TTDpdNDF, % pdNDF	80.44 ^a^	76.65 ^b^	79.39 ^a^	80.11 ^a^	1.03	0.01

^a,b^ Values within rows with different superscripts differ (*p* ≤ 0.05). ^1^ CTR—control; WDGS—condensed wheat distiller soluble diet; FF—former foodstuff; WDGS + FF—condensed wheat distiller soluble + former foodstuff. ^2^ CP—crude protein; aNDFom—amylase-treated ash-corrected neutral detergent fibre (NDF) with addition of sodium sulphite; ADF—acid detergent fibre; ADL—acid detergent lignin; uNDF240—undigested NDF at 240 h in vitro fermentation; pdNDF24—potentially digestible NDF; TTDpdNDF, % pdNDF—total tract digestibility of pdNDF.

## Data Availability

The data presented in this study are available on request from the corresponding author.

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
