# Peer review of "Combined Inclusion of Former Foodstuff and Distiller Grains in Dairy Cows Ration: Effect on Milk Production, Rumen Environment, and Fiber Digestibility"

_animals, 2022, doi:10.3390/ani12243519_

Round 1

Reviewer 1 Report

The research is very interesting and valuable. The introduction is exhaustive, the experimental design is properly planned. The results are clear and adequately presented. Before publishing in Animals, only slight corrections would be required.

Specific concerns:

Line 180: Add additional information about the methods used to VFA analyses.

Table 2 should be reported before then Table 1.

Tables 3 and 4: Lowercase letter "c" can be deleted since it is not reported in Table.

Table 6: a,bValues within rows with different superscripts differ (P ≤ .05) has been omitted.

Author Response

Dear Reviewer,

Thank you very much for your suggestions. We have modified the paper according to your indications.

Line 180: Add additional information about the methods used to VFA analyses.

AU: this section has been improved by adding detailed methodologies about VFA determination (L 151-157)

Table 2 should be reported before then Table 1.

AU. we have changed the order of these tables and then we've moved them to the M&M section, as suggested by reviewer 2.

Tables 3 and 4: Lowercase letter "c" can be deleted since it is not reported in Table.

AU deleted

Table 6: a,bValues within rows with different superscripts differ (P ≤ .05) has been omitted.

AU added

Reviewer 2 Report

Dear authors,

I have only few comments:

Chapter 2.1. – What breed of dairy cows was used in this study? Add this information in to the manuscript.

The aim of this study was to determine the effect of use of bakery former food products and wheat distillers grain with solubles on rumen environment, production and … of lactating dairy cows. There is no mention of detection of nutritional value of former food products and wheat distillers grain with solubles (even though this parameters are important for correct cows diet composition). My suggestion is to replace Table 1. and Table 2. to the chapter 2. Material and Methods. Then, “Results” will be started with description of Table 3.

Line 229-231 – Concentration of ammonia, total VFA and acetic acid were lower in WDGS compared only to CTR and FF. Not compared to all other (see superscripts in Table 4.).

Line 232 – Propionic acid was higher in FF only compared to WDGS (see superscripts in Table 4.).

Control and correct description of all significant differences exactly according to superscripts in tables 4. and 5.

Line 280 – De Boever et al. – this reference has number 48.

Line 296 – Karlsson et al. - this reference has number 49.

Control and correct all author citations in the text and in References.

Author Response

Dear Reviewer,

thank you very much for your suggestions. We fully agree with your indications and we have modified the paper accordingly. Please find below all details.

Chapter 2.1. – What breed of dairy cows was used in this study? Add this information in to the manuscript.

Au Holsteins, added. L52

The aim of this study was to determine the effect of use of bakery former food products and wheat distillers grain with solubles on rumen environment, production and … of lactating dairy cows. There is no mention of detection of nutritional value of former food products and wheat distillers grain with solubles (even though this parameters are important for correct cows diet composition). My suggestion is to replace Table 1. and Table 2. to the chapter 2. Material and Methods. Then, “Results” will be started with description of Table 3.

AU: We have specified that analysis of the experimental feeds have been performed before the trial, to formulate the rations proprerly. In addition, we moved the tables 1 and 2 in the M&M section, and accordingly to reviewer 1 we inverted their order.

Line 229-231 – Concentration of ammonia, total VFA and acetic acid were lower in WDGS compared only to CTR and FF. Not compared to all other (see superscripts in Table 4.).

AU. We specified better all statistical differences accordingly to superscript letters. (L190)

Line 232 – Propionic acid was higher in FF only compared to WDGS (see superscripts in Table 4.).

Control and correct description of all significant differences exactly according to superscripts in tables 4. and 5.

AU We specified better all statistical differences accordingly to superscript letters in both tables. (L190-192)

Line 280 – De Boever et al. – this reference has number 48.

Au corrected.

Line 296 – Karlsson et al. - this reference has number 49.

AU corrected

Control and correct all author citations in the text and in References.

Au We've checked and corrected all References.